# Risk of Psoriasis in Patients with Polycystic Ovary Syndrome: A National Population-Based Cohort Study

**DOI:** 10.3390/jcm9061947

**Published:** 2020-06-22

**Authors:** Tsung-Hsien Lee, Cheng-Hsuan Wu, Ming-Li Chen, Hei-Tung Yip, Chun-I Lee, Maw-Sheng Lee, James Cheng-Chung Wei

**Affiliations:** 1Institute of Medicine, Chung Shan Medical University, Taichung 40203, Taiwan; jackth.lee@gmail.com (T.-H.L.); 97528@cch.org.tw (C.-H.W.); adoctor0402@gmail.com (C.-I.L.); msleephd@gmail.com (M.-S.L.); 2Department of Obstetrics and Gynecology, Chung Shan Medical University Hospital, Taichung 40203, Taiwan; 3Division of Infertility, Lee Women’s Hospital, Taichung 40602, Taiwan; 4Women’s Health Research Laboratory, Changhua Christian Hospital, Changhua 50006, Taiwan; 5School of Medicine, Chung Shan Medical University, Taichung 40203, Taiwan; estellachen2939@gmail.com; 6School of Medicine, Kaohsiung Medical University, Kaohsiung 80708, Taiwan; 7Management Office for Health Data (DryLab), Clinical Trial Research Center, China Medical University Hospital, Taichung 40402, Taiwan; fionyip0i0@gmail.com; 8Division of Allergy, Immunology and Rheumatology, Chung Shan Medical University Hospital, Taichung 40203, Taiwan; 9Graduate Institute of Integrated Medicine, China Medical University, Chinese Medicine Clinical Trial Center, Chung Shan Medical University Hospital, Taichung 40402, Taiwan

**Keywords:** polycystic ovary syndrome, psoriasis, metabolic syndrome, National Health Insurance Research Database

## Abstract

Both polycystic ovary syndrome (PCOS) and psoriasis are associated with insulin resistance and metabolic syndrome. Nonetheless, the incidence of psoriasis in patients with PCOS is unclear. We used the Longitudinal Health Insurance Research Database (LHID) in Taiwan from 2000 to 2012 to perform a retrospective population-based cohort study to elucidate the occurrence of psoriasis in PCOS patients. Patients with PCOS without psoriasis in the index year (the year of PCOS diagnosis) were recruited as the PCOS group. Those without PCOS nor psoriasis (control group) were selected using propensity score matching at a ratio of 4:1. Hazard ratios (HRs) were obtained using the Cox proportional hazards regression model. In total, 4707 and 18,828 patients were included in the PCOS and control groups, respectively. The incidence rates of psoriasis in the control and PCOS groups were 0.34 and 0.70 per 1000 person-years, respectively. The risk of psoriasis was higher in the PCOS group by an HR of 2.07 (95% confidence interval [CI] = 1.25–3.43) compared with the control group. In conclusion, the incidence of psoriasis in the PCOS group was higher than that in the control group. Further studies should be conducted to investigate the mechanism underlying the association, and to benefit the long-term management of patients with PCOS.

## 1. Introduction

Polycystic ovary syndrome (PCOS) is a chronic anovulation disorder, and it is diagnosed if two of three diagnostic criteria are met: oligomenorrhea, hyperandrogenism, and polycystic ovarian morphology [1]. Elevated androgen and insulin resistance are key pathologic features of PCOS [2]. Obesity is common in PCOS patients and correlates to insulin resistance [2]. Furthermore, elevated androgen may further contribute insulin resistance [3]. Hyperinsulinemia, dyslipidemia and dysglycemia (fasting glucose > 100 mg/dL) resulted from insulin resistance in PCOS patients [3]. Such metabolic effects composed metabolic syndrome in adults. Consequently, insulin resistance connects PCOS with metabolic syndrome; thus, patients with PCOS are predisposed to develop metabolic syndrome in their 40s [4].

Psoriasis is a chronic skin inflammatory disease, featuring scaly and erythematous plaques. The prevalence of psoriasis varies geographically; as per the medical literature, its incidence is approximately 0.6–6.5% in Europe and 3% in the US [5]. The etiology of psoriasis is unknown; nonetheless, it is a complex disease believed to result from the interaction of genes, the immune system, and even the endocrine system. Moreover, evidence has indicated that psoriasis is closely associated with insulin resistance [6,7,8] and metabolic syndrome [9,10].

Patients with psoriasis were reported to be at risk of PCOS in a cross-sectional study in Italy [7], as both PCOS and psoriasis are correlated with hyperandrogenism and metabolic syndrome. In addition, psoriatic patients with PCOS have more severe skin lesions compared to those without PCOS [11]. However, the risk of psoriasis in PCOS patients is largely unknown. Since insulin resistance and metabolic syndrome are common features to PCOS and psoriasis, we propose that patients with PCOS might have a high incidence of psoriasis. This study analyzed a population-based cohort database to prove that patients with PCOS have a higher chance of developing psoriasis in later life than patients without PCOS.

## 2. Experimental Section

### 2.1. Data Source

The National Health Insurance (NHI) program contains the medical records of almost the entire population of Taiwan. It was established in 1995 when the Taiwanese government launched the NHI program. In this study, we used the Longitudinal Health Insurance Database (LHID), which is a subset of the medical records in the NHI program; the LHID includes the data of 1,000,000 randomly selected insurants. The database has been validated in many studies [12,13,14]. The LHID also contains outpatient visit and hospitalization records as well as data on medication usage and treatment. Complying with ethical guidelines, all identifying data were encrypted. This study was approved by the International Review Board (IRB) of China Medical University and Hospital (IRB permit number: CMUH-104-REC2–115). The diseases in the LHID were coded according to the International Classification of Diseases, Ninth Revision, Clinical Modification (ICD-9-CM). The present analysis is presented according to the guideline of STROBE statement for reporting a cohort study (Appendix A: STROBE cohort checklist).

### 2.2. Study Participants

We selected patients to be the case and control groups from the LHID containing 1,000,000 Taiwanese medical records in NIH program by the following strategy. Patients diagnosed as having PCOS (ICD-9-CM code 256.4) between 2000 and 2012 were recruited as the case group in this cohort study. The disease was identified by the presence of at least three related outpatient diagnoses or at least one related inpatient diagnosis to ensure the validity of disease diagnosis. Women with a history of psoriasis (ICD-9-CM codes 696.0 and 696.1) prior to the diagnosis of PCOS were excluded. The control group included women free of PCOS and psoriasis matched to the participants of the PCOS group by age and index year through propensity score matching at the ratio of 1:4. The index date for patients with PCOS was defined as the date of PCOS diagnosis. We excluded participants younger than 14 years and those with psoriasis diagnosed prior to the index year.

### 2.3. Main Outcome and Comorbidities

The study assessed the incidence of psoriasis in the PCOS and control groups. We considered numerous related comorbidities including anxiety (ICD-9-CM code 493), asthma (ICD-9-CM code 493), coronary artery disease (CAD; ICD-9-CM codes 410–413, 414.01–414.05, 414.8, and 414.9), cancer (ICD-9-CM codes 140–208), chronic obstructive pulmonary disease (COPD; ICD-9-CM codes 490–496), congestive heart failure (CHF; ICD-9-CM codes 398.91, 402.01, 402.11, 402.91, and 428), chronic liver diseases (CLD; ICD-9-CM code 571.4), diabetes mellitus (DM; ICD-9-CM code 250), depression (ICD-9-CM codes 296.2, 296.3, 296.5, 296.82, 300.4, and 309.0–309.1), hypertension (ICD-9-CM codes 401–405), hyperlipidemia (HLA; ICD-9-CM code 272), stroke (ICD-9-CM codes 430–438), and sleep apnea (ICD-9-CM codes 327.2, 780.51, 780.53, and 780.57).

### 2.4. Statistical Analysis

The chi-square test was conducted to examine the differences in categorical variables between the control and PCOS groups. The statistical methods for PCOS and other diseases in NHID have been validated in previous reports [12,13,14]. The mean ages were compared using Student’s *t*-test. We obtained the cumulative incidence curve using the Kaplan–Meier method and assessed the difference between the groups using the log-rank test. Hazard ratios (HRs) were obtained using the Cox proportional hazards regression model. We also performed stratification analysis to elucidate the interaction between PCOS and comorbidities to investigate if it influences the incidence of psoriasis in the case and control groups. All statistical analyses were performed by using SAS version 9.4 (Version 9.4, SAS Institute Inc., Cary, NC, USA). The significance level for statistical analysis was set at *p* < 0.05.

## 3. Results

During the follow-up period, 4707 patients with PCOS and 18,828 controls were identified from the LHID. The mean follow-up times of the control and PCOS groups were 6.94 ± 3.53 and 6.99 ± 3.53 years, respectively. Table 1 shows the distribution of the baseline characteristics of both groups. The age of both groups did not differ significantly (27.2 ± 6.68 vs. 27.3 ± 6.95, *p* > 0.05). The PCOS group had a significantly higher proportion of comorbidities, such as asthma, COPD, CLD, DM, hypertension, HLA, depression, and sleep apnea, than the control group.

The cumulative incidence curve of psoriasis was higher in the PCOS group than in the control group (*p* = 0.004, log-rank test, Figure 1). Table 2 demonstrates the association between psoriasis and explanatory variables. The incidence rates of psoriasis in the control and PCOS groups were 0.34 and 0.70 per 1000 person-years, respectively. The PCOS group showed a higher risk of psoriasis by an HR of 2.07 (95% confidence interval [CI] = 1.25–3.43, *p* < 0.01) compared with the control group. People aged more than 50 years were more likely to develop psoriasis by an HR of 14.13 (95% CI = 1.8–110.7, *p* < 0.05) relative to the population aged less than 20 years. Moreover, cancer raised the risk of psoriasis by an HR of 11.72 (95% CI = 2.87–47.9, *p* < 0.001) compared with those without cancer.

Table 3 demonstrates the HRs of psoriasis in the PCOS and control groups in analysis stratified by various comorbidities. Notably, we found that the risk of psoriasis in the PCOS group was more prominent at a young age (<20 years, HR: 4.02, 95% CI = 1.16–13.9, *p* < 0.05; 20–50 years, HR: 1.88, 95% CI = 1.07–3.29, *p* < 0.05). Cancer and PCOS are independent risk factors for psoriasis because the HR of psoriasis in the PCOS group did not change in cancer-stratified analysis. Moreover, in the analysis stratified by comorbidities, such as asthma, COPD, CLD, DM, hypertension, HLA, depression, and sleep apnea, the HR of psoriasis in the PCOS group did not change significantly.

## 4. Discussion

Our study analyzed a population-based database and demonstrated that the PCOS group had a higher chance of developing comorbidities, such as asthma, COPD, CLD, DM, hypertension, HLA, and depression, compared with the control group. Among these comorbidities, DM, hypertension, and HLA represented the late consequences of metabolic syndrome. The PCOS group did not show a high risk of CAD and stroke in the present study. These results suggest that compared with the control group, the PCOS group had a higher predisposition to most diseases related to metabolic syndrome or insulin resistance, except two vasculature-related abnormalities, namely CAD (0.85% vs. 0.65%, respectively, *p* = 0.162) and stroke (0.68% vs. 0.59%, *p* = 0.573).

In a recent study using the Taiwanese LHID to study the association between PCOS and risk of depression, the PCOS group again did not have CAD risk (0.8% vs. 0.6%, *p* = 0.1421) compared with the non-PCOS group [15]. It also revealed that the association between PCOS and CAD became evident only after a follow-up time of more than 2 years at least [16]. CAD usually occurs later in life among women than in men, whereas PCOS is diagnosed at an early reproductive age. This may explain the nonsignificant difference of CAD events between the PCOS and control groups in the present study regarding to the incidence of psoriasis.

Stroke is another metabolic syndrome-related comorbidity with vasculature-related abnormalities. Although previous reports of PCOS using the LHID in Taiwan demonstrated a higher incidence of stroke in patients with PCOS (1.5% vs. 0.9%, *p* < 0.001) [15] and (1.6% vs. 1.0%, *p* < 0.001) [3], our study showed no significant difference between the PCOS and control groups. A recent meta-analysis provided conflicting results for the risk of stroke in patients with PCOS [17] and showed that the association between PCOS and stroke became nonsignificant after adjustment for the body mass index [17]. In the present study, we did not stratify the PCOS patients with obesity (body mass index. Therefore, obesity may be a potential confounder and consequently results in such observation. Another possible explanation for non-correlation of PCOS with CAD and stroke in the present study is that psoriasis itself is associated with an elevated risk of stroke and CAD. In a Danish nationwide study, the risks of stroke and myocardial infarction (MI) increased by incidence rate ratios of 1.95 (95% CI 1.43–2.66) and 1.57 (95% CI 1.07–2.29), respectively, in patients with psoriasis [18]. Because we excluded patients with psoriasis in the index year for both groups, the prevalence rates of stroke in PCOS patients in our study (0.68%) are less than half of those in the two previous reports (1.5% and 1.6%) [3,15]. Therefore, in the present study, stroke and CAD risks in the PCOS group might be attenuated under the exclusion of psoriatic patients diagnosed before the index year.

In the present analysis, three main risk factors (advanced age, PCOS, and cancer) were identified by adjusting HRs using the Cox regression model, suggesting that they are substantial risk factors for psoriasis. Psoriasis might occur at any age, notably peaking at approximately 20–30 and over 50 years of age [19] because of hormonal changes at puberty and menopause, which are the reported trigger factors [19]. Nonetheless, the data in the present study demonstrated that the prevalence of psoriasis is higher in the advanced-age population (>50 years of age) than in the young population (<20 years of age).

Using both the Cox regression model and stratification analysis, our study also revealed that cancer and PCOS are independent factors for the incidence of psoriasis. Based on LHID analysis, patients with psoriasis were reported to have a higher risk of cancer than people without psoriasis [20]. A meta-analysis also indicated a small increase in the risk of cancer among patients with psoriasis [21]. Psoriasis is an immune-mediated disease characterized by dysregulation of several cytokines, such as tumor necrosis factor-alpha (TNF-α). Therefore, systemic inflammation plays a significant role in the pathogenesis of psoriasis [22]. Moreover, patients with psoriasis are at a high risk of human immunodeficiency virus (HIV) or human papilloma virus (HPV) infection as per recent reports based on the LHID in Taiwan [23,24]. Hence, the cancer risk among patients with psoriasis might be related to such carcinogenic viral infections in addition to immune dysregulation. Although patients with PCOS also demonstrate a low grade of inflammation subsequent to hyperandrogenism and IR [25], the immune dysfunction in patients with psoriasis with oncogenic virus infection might be more complicated than that in patients with PCOS.

In a previous cross-sectional analysis, the psoriasis and control groups showed PCOS prevalence rates of 47% and 11%, respectively [7]. Moreover, hypertension and metabolic syndrome did not change the odds ratio of PCOS prevalence [7]. Our study showed a higher cumulative incidence rate of psoriasis in the PCOS group than in the control group, with an HR of 2.07 for the PCOS group, which remained unaffected by hypertension, DM, and HLA. Both reports indicated the close correlation between psoriasis and PCOS. The cumulative incidence of psoriasis in patients with PCOS increased gradually with the follow-up time in the present study. Insulin resistance is the common pathologic mechanism in these two diseases. Taken together, our results indicate that PCOS might be an earlier IR event than metabolic syndrome and psoriasis. Nonetheless, the skin lesions are more severe in psoriasis combined with PCOS compared to those in psoriasis alone [11]. Whether control of insulin resistance in PCOS patients would decrease the incidence or severity of psoriasis deserves further investigation.

The strength of the present study is the application of a nationwide population-based dataset. The NHI program covers almost the whole population of Taiwan. We used the longitudinal data of 1,000,000 people randomly selected from the NHI program between 2000 and 2012. The dataset allowed us to analyze the incidence of relatively rare diseases with a mean follow-up period of approximately 6 years, which was helpful to elucidate the causal relationship between two diseases. Such a population-based retrospective cohort analysis is an economical and suitable alternative to the more expensive prospective population-based cohort study.

The use of the LHID, however, is associated with some limitations in the present study. First, a misclassification bias might result from the coding process by each physician and hospital in Taiwan for health insurance claims. The NHI Administration in Taiwan consistently audits ICD-M-9 codes in every medical institute and clinic. Therefore, the bias would be minimal in the LHID. In addition, we used at least three consecutive clinic visits for the same ICD-M-9 code to increase diagnostic validity. Consequently, the diagnosis of PCOS and psoriasis is validated and this further eliminates the misclassification bias. Second, data on potential confounding factors, such as smoking, alcohol consumption, and socioeconomic status, are not available in the LHID and could not be analyzed in the present study. Nonetheless, COPD could be the proxy variable for smoking, which was adjusted and did not affect the HR of psoriasis in the present study. Third, detailed data about body mass index in the index year are not available. The confounding effect of obesity is not minimized in the present study. Forth, the LHID only recruited Taiwan people, and ethnicity is a confounding factor for the incidence of psoriasis. The results in the present study need be carefully interpreted to apply directly for the people living at other geographic regions.

In conclusion, our study showed that the incidence of psoriasis is higher in the PCOS group than in the control group. The younger the age of PCOS diagnosis, the higher the cumulative incidence of psoriasis. The comorbidities associated with insulin resistance or metabolic syndrome, such as DM, hypertension, HLA, CAD, and stroke, did not modify the HR of psoriasis in patients with PCOS. Further mechanism studies regarding insulin resistance are required for the long-term management of PCOS patients.

## Figures and Tables

**Figure 1 jcm-09-01947-f001:**
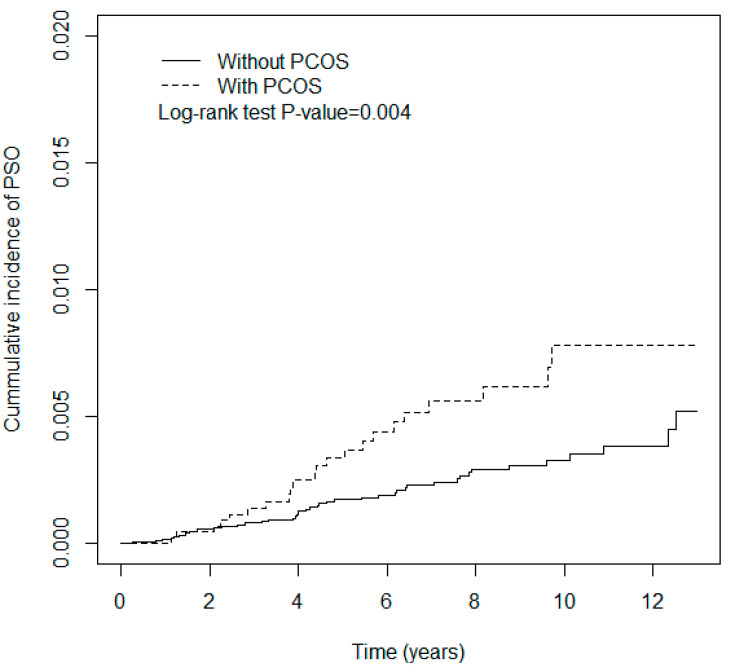
Cumulative incidence of psoriasis for patients with and without PCOS.

**Table 1 jcm-09-01947-t001:** Baseline characteristics and comorbidities of patients with and without polycystic ovary syndrome (PCOS).

	PCOS				
	No (*N* = 18,828)		Yes (*N* = 4707)		
Variables	n	%	n	%	*p*-Value ^1^
Age, year					>0.99
≤20	2791	15	697	15	
20–50	16,017	85	4005	85	
>50	20	0.11	5	0.11	
Comorbidities					
Asthma	916	4.9	316	6.7	<0.001
Chronic obstructive pulmonary diseases	2064	11	660	14	<0.001
Coronary arterial diseases	122	0.65	40	0.85	0.162
Cancer	63	0.33	14	0.30	0.797
Congestive heart failure	40	0.21	8	0.17	0.691
Chronic liver diseases	948	5.0	375	8.0	<0.001
Diabetes Mellitus	261	1.4	143	3.0	<0.001
Hypertension	290	1.5	111	2.4	<0.001
Hyperlipidemia	464	2.5	254	5.4	<0.001
Stroke	112	0.59	32	0.68	0.573
Anxiety	52	0.28	20	0.42	0.132
Depression	740	3.9	252	5.4	<0.001
Sleep apnea	19	0.10	11	0.23	0.040

^1^*p* value by chi-square test.

**Table 2 jcm-09-01947-t002:** Incidence rates and hazard ratios of psoriasis.

	Psoriasis						
Variables	Events	PY	IR	Crude HR	(95%CI)	Adjusted HR ^ꝉ^	(95%CI)
PCOS							
No	44	130,684	0.34	1.00	-	1.00	-
Yes	23	32,904	0.70	2.08	(1.25,3.44) **	2.07	(1.25,3.43) **
Age, year							
≤20	10	24,622	0.41	1.00	-	1.00	-
20–50	56	138,783	0.40	1.00	(0.51,1.95)	0.98	(0.5,1.93)
>50	1	183	5.45	13.82	(1.76,108.26) *	14.13	(1.8,110.7) *
Comorbidities							
anxiety							
No	67	163,175	0.41	1.00	-		
Yes	0	413	0	0.00	(0, Inf)		
asthma							
No	64	156,490	0.41	1.00	-		
Yes	3	7098	0.42	1.07	(0.34,3.4)		
CAD							
No	67	162,514	0.41	1.00	-		
Yes	0	1074	0	0.00	(0,Inf)		
cancer							
No	65	163,144	0.40	1.00	-	1.00	-
Yes	2	444	4.50	11.70	(2.86,47.83) ***	11.72	(2.87,47.9) ***
COPD							
No	60	147,133	0.41	1.00	-		
Yes	7	16,455	0.43	1.07	(0.49,2.35)		
CHF							
No	67	163,310	0.41	1.00	-		
Yes	0	278	0	0.00	(0,Inf)		
CLD							
No	62	154,714	0.40	1.00	-		
Yes	5	8874	0.56	1.42	(0.57,3.54)		
DM							
No	65	160,929	0.40	1.00	-		
Yes	2	2659	0.75	1.88	(0.46,7.69)		
Depression							
No	65	157,679	0.41	1.00	-		
Yes	2	5909	0.34	0.84	(0.21,3.44)		
Hypertension							
No	67	161,058	0.42	1.00	-		
Yes	0	2531	0	0.00	(0,Inf)		
Hyperlipidemia							
No	64	159,002	0.40	1.00	-		
Yes	3	4586	0.65	1.65	(0.52,5.24)		
Stroke							
No	66	162,624	0.41	1.00	-		
Yes	1	965	1.04	2.56	(0.35,18.43)		
Sleep apnea							
No	67	163,446	0.41	1.00	-		
Yes	0	142	0	0.00	(0,Inf)		

*: *p* value < 0.05; **: *p* value < 0.01; ***: *p* value < 0.001; PY: person-years; IR: incidence rate (per 1000 person-years); CAD: coronary artery diseases; COPD: chronic obstructive pulmonary diseases; CHF: congestive heart failure; CLD: chronic liver diseases; DM: diabetes mellitus; ^ꝉ^: adjusted for age and cancer.

**Table 3 jcm-09-01947-t003:** Association of PCOS and psoriasis at different stratification levels.

	PCOS							
	No			Yes				
Variables	Events	PY	IR	Events	PY	IR	Adjusted HR ^ꝉ^	(95%CI)
Age, year								
≤20	5	19,714	0.25	5	4908	1.02	4.02	(1.16,13.9) *
20–50	38	110,823	0.34	18	27,960	0.64	1.88	(1.07,3.29) *
>50	1	148	6.78	0	36	0	-	-
Comorbidities								
anxiety								
No	44	130,380	0.34	23	32,795	0.70	2.07	(1.25,3.43) **
Yes	0	304	0	0	109	0	-	-
asthma								
No	43	125,433	0.34	21	31,058	0.68	1.96	(1.16,3.30) *
Yes	1	5251	0.19	2	1847	1.08	5.62	(0.51,62.4)
CAD								
No	44	129,855	0.34	23	32,659	0.70	2.07	(1.25,3.43) **
Yes	0	829	0	0	245	0	-	-
cancer								
No	43	130,330	0.33	22	32,814	0.67	2.03	(1.21,3.40) **
Yes	1	354	2.83	1	91	11.05	1.44	(0.26,68.2)
COPD								
No	40	118,275	0.34	20	28,859	0.69	2.03	(1.18,3.47) **
Yes	4	12,409	0.32	3	4046	0.74	2.27	(0.57,10.2)
CHF								
No	44	130,439	0.34	23	32,871	0.70	2.07	(1.25,3.43) **
Yes	0	245	0	0	33	0	-	-
CLD								
No	42	124,509	0.34	20	30,205	0.66	1.96	(1.15,3.34) *
Yes	2	6175	0.32	3	2699	1.11	2.78	(0.46,16.9)
DM								
No	43	128,984	0.33	22	31,945	0.69	2.05	(1.23,3.43) **
Yes	1	1700	0.59	1	959	1.04	1.75	(0.11,28.0)
Depression								
No	42	126,183	0.33	23	32,859	0.70	2.18	(1.31,3.62) **
Yes	2	4501	0.44	0	45	0	-	-
Hypertension								
No	44	128,876	0.34	23	32,182	0.71	2.07	(1.25,3.44) **
Yes	0	1808	0	0	722	0	-	-
Hyperlipidemia								
No	43	127,796	0.34	21	31,206	0.67	1.99	(1.18,3.36) **
Yes	1	2888	0.35	2	1698	1.18	3.09	(0.27,34.1)
Stroke								
No	44	129,947	0.34	22	32,677	0.67	1.98	(1.19,3.30) **
Yes	0	737	0	1	227	4.40	-	-
Sleep apnea								
No	44	130,587	0.34	23	32,859	0.70	2.07	(1.25,3.43) **
Yes	0	97	0	0	45	0	-	-

*: *p* value < 0.05; **: *p* value < 0.01; ***: *p* value < 0.001; PY: person-years; IR: incidence rate (per 1000 person-years); CAD: coronary artery diseases; COPD: chronic obstructive pulmonary diseases; CHF: congestive heart failure; CLD: chronic liver diseases; DM: diabetes mellitus; ^ꝉ^: adjusted for age and cancer.

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
