# Peer review of "Risk of Psoriasis in Patients with Polycystic Ovary Syndrome: A National Population-Based Cohort Study"

_jcm, 2020, doi:10.3390/jcm9061947_

Round 1

Reviewer 1 Report

This is a well written paper that provides us with interesting “real life” data.

I enjoyed reading the paper however it lacks the proper logistic regression based on personal demographics. I do urge the authors to assess whether the BMI of patients is the missing link. We know that patients with polycystic ovaries have a higher BMI which is linked with the appearance of psoriasis. This analysis should be performed as well as the patients’’ age.  

Author Response

This is a well written paper that provides us with interesting “real life” data. I enjoyed reading the paper however it lacks the proper logistic regression based on personal demographics. I do urge the authors to assess whether the BMI of patients is the missing link. We know that patients with polycystic ovaries have a higher BMI which is linked with the appearance of psoriasis. This analysis should be performed as well as the patients’’ age.

Response:
Thank you for the precious comments.
Since the detail data of body mass index is not available in the database from Taiwan Health Insurance Program, we are not able to analysis the confounding effect of BMI or obesity in the present study. The
confounding bias by obesity is discussed at t he limitation section.
The present study used the Longitudinal Health Insurance Database (LHID) from 2000 to 2012 for comparing the occurrence of psoriasis in the PCOS and control groups. Therefore, we used Cox regression model to analyze this time-event data to obtain the adjusted hazard ratio. The log-rank test and Kaplan–Meier method are performed as sensitivity analysis.

Reviewer 2 Report

This is an potentially very interesting study, but I believe that the paper can be greatly improved.

The study design is appropiate, but patients in the PCOS and non-PCOS group are matched by age and index year and not by weight. Obesity has been demostrated to be asociated to both PCOS and psoriasis, and could act as a counfounding bias in this study. This must be mentioned in the limitations section of the paper.

Discussion is the main part of the paper that has to be improved. The way it is written is very confusing. While the main objective of the paper is "to prove that patients with PCOS have a higher chance of developing psoriasis in later life than patients without PCOS", it begins discussing its results on the relationship of PCOS and its comorbidities, excluding psoriasis. I believe this section should be completely rewritten, giving the association between PCOS and poriasis much more importance than other secondary objectives of the study.

English could also be improved; "in the Europe" (line 53), "believed to resulting" (line 54) are incorrect, and in line 150 "suggest" would read better than "suggested"

Author Response

This is an potentially very interesting study, but I believe that the paper can be greatly improved. The study design is appropiate, but patients in the PCOS and non-PCOS group are matched by age and index year and not by weight. Obesity has been demostrated to be asociated to both PCOS and psoriasis, and could act as a counfounding bias in this study. This must be mentioned in the limitations section of the paper.

Response: The confounding bias by obesity is discussed at the limitation section.

Discussion is the main part of the paper that has to be improved. The way it is written is very confusing. While the main objective of the paper is "to prove that patients with PCOS have a higher chance of developing psoriasis in later life than patients without PCOS", it begins discussing its results on the relationship of PCOS and its comorbidities, excluding psoriasis. I believe this section should be completely rewritten, giving the association between PCOS and poriasis much more importance than other secondary objectives of the study.

Response: Thank you for the valuable comments.

In the section of Discussion, we attempted initially to figure out why the correlation of PCOS with CAD and stroke is absent. Therefore, we discussed the relationship of CAD and stroke to psoriasis. Then we suggest that the exclusion criteria in the present PCOS- control study may lead to this observation.

Then we discussed the relationship between cancer and psoriasis, PCOS about the common feature in immune dysfunction and low-grade inflammation. This part indicated that immune or inflammation may also play a role in the occurrence of psoriasis in PCOS patients.

We provide additional information about the impact of psoriasis with PCOS vs. psoriasis without PCOS on skin lesion severity in the subsequent discussion.

All the discussions are surround the PCOS and psoriasis. We tried to explain the multiple perspectives of correlation between PCOS and psoriasis.

English could also be improved; "in the Europe" (line 53), "believed to resulting" (line 54) are incorrect, and in line 150 "suggest" would read better than "suggested"

Response: We corrected the English according to reviewer’s comments.

Reviewer 3 Report

Thank you for the opportunity to review this paper, manuscript jcm-814249. The aim of this paper was to examine the relationship between PCOS and psoriasis using the Longitudinal Health Insurance Research Database in Taiwan.

Overall comment:

The idea and aim of this paper is valid and very much needed, however additional information needs to be added to the paper (e.g. clearer information outlining information on the statistical analysis). The overall structure of the paper is coherent but the discussion does need to be improved. The idea of each discussion paragraph needs to be identified. The flow from one paragraph to the next also may be improved. This will make it much easier for the reader to understand this paper. Particular attention to sentence structure is also required. To make the study clearer it is recommended authors complete the STROBE cohort checklist, edit the paper accordingly according to the STROBE and upload it as supplementary data.

These are additional comments for the authors to consider:

Abstract

Suggestions:

Make it clear that controls had no psoriasis or PCOS. Suggest rewording line 31-32.

Line 30: outline which association you are referring to, to make it clearer to the reader.

Introduction

Please provide additional information relating IR to PCOS and how this relates to Metabolic Syndrome. The current rationale is a bit too brief. Bit more information rather than ‘IR connects to Metabolic Syndrome’ is recommended.

Provide clearer gaps in research and how this study aims to contribute to these gaps. Yu outline on line 58 that patients with psoriasis have been reported to be at risk of PCOS but the age, country, ethnicity are not outlined. Any other studies relating to psoriasis and PCOS?

Clearly define the aim and hypothesis as the end of the introduction to make it clear for the reader to understand.

Experimental section

Study participants

Under study participants is it recommended that the inclusion and exclusion criteria of participants is clearly outlined.

Suggest making clear that the control group had neither PCOS nor psoriasis on line 80.

Statistical analysis

It is recommended to please outline further details of your statistical methods:

e.g. statistical package used, normality testing and how this was completed, categorisation of variables, significance levels. Suggest adding references to validate statistical methods. Did authors complete any tests to exclude or assess for bias?

Results

To make it clear for the reader it is recommended that p-values are not included in the results section and kept to that only outlined in Table 1.

Formatting of Table 1 is recommended – e.g. remove % from within table, add abbreviations below table, briefly add methods use to assess data outlined in Table 1 as footnotes.

Discussion

A good discussion with some great points. It is recommended that authors remove lines 195-199 as this is restating the methods. What are some reasons that the authors suggest may link stroke and PCOS? Providing reasons for the link in addition to previous studies researching the association is beneficial.

Please add references throughout the paper when you make claims e.g. at the end of line 194.

Please move exclusion information on 178 and 179 to the methods study participants section.

One line 221 authors outline that 3 consecutive clinic visits were used to increase diagnostic validity. It is presumed that this is a strength of the study. Suggest rewording so this is made clearer.

Author Response

Thank you very much for the valuable comments. Please see our responses in the following paragraph in italic style.

Overall comment:

The idea and aim of this paper is valid and very much needed, however additional information needs to be added to the paper (e.g. clearer information outlining information on the statistical analysis). The overall structure of the paper is coherent but the discussion does need to be improved. The idea of each discussion paragraph needs to be identified. The flow from one paragraph to the next also may be improved. This will make it much easier for the reader to understand this paper. Particular attention to sentence structure is also required. To make the study clearer it is recommended authors complete the STROBE cohort checklist, edit the paper accordingly according to the STROBE and upload it as supplementary data.

Response:  Thank you for the valuable comments. The discussion is reworded to make the idea in each discussion paragraph clearer. We complete the STROBE cohort checklist and upload it as supplementary data.

These are additional comments for the authors to consider:

Abstract

Suggestions:

Make it clear that controls had no psoriasis or PCOS. Suggest rewording line 31-32.

Line 30: outline which association you are referring to, to make it clearer to the reader.

Response:  The line 30, 31-32 was changed to be the following: We used the Longitudinal Health Insurance Research Database (LHID) in Taiwan from 2000 to 2012 to perform a retrospective population-based cohort study to elucidate the occurrence of psoriasis in PCOS patients. Patients with PCOS without psoriasis in the index year (the year of PCOS diagnosis) were recruited as the PCOS group. Those without PCOS nor psoriasis (control group) were selected using propensity score matching at a ratio of 4:1.

Introduction

Please provide additional information relating IR to PCOS and how this relates to Metabolic Syndrome. The current rationale is a bit too brief. Bit more information rather than ‘IR connects to Metabolic Syndrome’ is recommended.

Provide clearer gaps in research and how this study aims to contribute to these gaps. Yu outline on line 58 that patients with psoriasis have been reported to be at risk of PCOS but the age, country, ethnicity are not outlined. Any other studies relating to psoriasis and PCOS?

Clearly define the aim and hypothesis as the end of the introduction to make it clear for the reader to understand.

Response: Thank you for the precious comments. We provide additional information about insulin resistance and the subsequent metabolic syndrome in PCOS patients. We also further define the aim and hypothesis of this study at the end of the introduction.  

Experimental section

Study participants

Under study participants is it recommended that the inclusion and exclusion criteria of participants is clearly outlined.

Suggest making clear that the control group had neither PCOS nor psoriasis on line 80.

Response: We provide additional information about the inclusion and exclusion criteria of participants as the following: “We selected patients to be the case and control groups from the LHID containing 1,000,000 Taiwanese medical records in NIH program by the following strategy. Patients diagnosed as having PCOS (ICD-9-CM code 256.4) between 2000 and 2012 were recruited as the case group in this cohort study. The disease was identified by the presence of at least three related outpatient diagnoses or at least one related inpatient diagnosis to ensure the validity of disease diagnosis. Women with a history of psoriasis (ICD-9-CM codes 696.0 and 696.1) prior to the diagnosis of PCOS were excluded. The control group included women free of PCOS and psoriasis matched to the participants of the PCOS group by age and index year through propensity score matching at the ratio of 1:4. The index date for patients with PCOS was defined as the date of PCOS diagnosis. We excluded participants younger than 14 years and those with psoriasis diagnosed prior to the index year.”

Statistical analysis

It is recommended to please outline further details of your statistical methods:

e.g. statistical package used, normality testing and how this was completed, categorisation of variables, significance levels. Suggest adding references to validate statistical methods. Did authors complete any tests to exclude or assess for bias?

Response: We provide further details about the statistical methods, including the statistical package, significance levels. The references about validating statistical methods are added.

Results

To make it clear for the reader it is recommended that p-values are not included in the results section and kept to that only outlined in Table 1.

Formatting of Table 1 is recommended – e.g. remove % from within table, add abbreviations below table, briefly add methods use to assess data outlined in Table 1 as footnotes.

Response: Thank you for the comments. We format Table1 according to reviewer’s comment.

Discussion

A good discussion with some great points. It is recommended that authors remove lines 195-199 as this is restating the methods. What are some reasons that the authors suggest may link stroke and PCOS? Providing reasons for the link in addition to previous studies researching the association is beneficial.

Please add references throughout the paper when you make claims e.g. at the end of line 194.

Please move exclusion information on 178 and 179 to the methods study participants section.

One line 221 authors outline that 3 consecutive clinic visits were used to increase diagnostic validity. It is presumed that this is a strength of the study. Suggest rewording so this is made clearer.

Response: Thank you for the valuable comments. We remove lines 195-199. In the present study, we attempted to figure out why the correlation of PCOS with CAD and stroke is absent. Therefore, we discussed the relationship of CAD and stroke to psoriasis. Then we suggest that the exclusion criteria in this PCOS- control comparing study may lead to this observation.

The end of line 194 was reworded to match the reference about the oncogenic virus infection and psoriasis.

Online 221, we provide additional information why we use three consecutive visits to avoid misclassification bias.

Round 2

Reviewer 1 Report

Accept as is

Author Response

Thank you for the precious comments.

Reviewer 2 Report

The changes have greatly improved the manuscript.

In lines 235 and 236, I consider "Nonetheless, skin lesions are more severe in psoriasis combined with PCOS compared to those in psoriasis alone" would read better.

Author Response

 Thank you for the valuable comments. We corrected the line 235-236 according to reviewer’s comment.